# Symptom Experiences and Coping Strategies in Turkish Patients with Implantable Cardioverter-Defibrillators: A Cross-Sectional Study Based on Interviews

**DOI:** 10.3390/healthcare14010059

**Published:** 2025-12-26

**Authors:** Sebiha Aktaş Us, Sultan Taşcı

**Affiliations:** 1Public Hospital Services Unit, Sinop Provincial Health Directorate, Turkish Ministry of Health, 57000 Sinop, Türkiye; 2Department of Internal Medicine Nursing, Faculty of Health Sciences, Erciyes University, 38260 Kayseri, Türkiye; sultant@erciyes.edu.tr

**Keywords:** implantable cardioverter defibrillator, symptom experience, spiritual coping, cultural context, Turkish patients

## Abstract

**Background/Objectives:** Although implantable cardioverter defibrillators (ICDs) play a critical role in reducing the risk of sudden cardiac death, patients may report various physical and psychological symptoms during the implantation process. This study aimed to examine ICD patients’ retrospective reports of pre-implantation symptoms, their concurrent assessment of post-implantation symptom experiences, and the non-pharmacological methods they used to cope with these symptoms. **Methods**: A cross-sectional study was conducted with ICD patients who visited the arrhythmia clinic between May and August 2022. Data were collected using a questionnaire developed by the researchers and medical records. The study analyzed changes in symptoms reported by patients, individual coping methods used, the perceived effectiveness of these methods, and comparisons of methods used according to descriptive and clinical characteristics. **Results**: Patients reported a decrease in chest pain, palpitations, dizziness, syncope, and shortness of breath after ICD implantation (*p* < 0.001). However, they reported an increase in fatigue and anxiety levels (*p* < 0.001); no significant change was reported for insomnia (*p* = 0.473) and fear (*p* = 0.082). Furthermore, no significant difference was observed between patients who received shock therapy and those who did not in terms of changes in anxiety and fatigue levels, and the increase in anxiety was similar in both groups. The most frequently reported coping method among patients was praying, followed by drinking herbal tea and walking. A significant relationship was found between marital status and coping method preference, while no relationship was found with other descriptive and clinical characteristics. **Conclusions**: Although ICD implantation reduces cardiac symptoms, anxiety and fatigue continue to increase. Patients’ tendency to turn to cultural and spiritual coping methods such as prayer indicates that post-ICD care should be conducted with a holistic approach that also covers psychological and spiritual needs.

## 1. Introduction

Sudden cardiac death (SCD) is death occurring within one hour of the onset of symptoms related to heart disease. It continues to be a serious public health problem on a global and national scale. Despite advances, SCD accounts for approximately half of all premature deaths related to heart disease [1]. In the US, it affects more than 600,000 people annually, with most cases occurring outside of hospitals [2,3]. The exact incidence rate in Türkiye is unknown [4,5].

The most common cause of SCD is life-threatening ventricular arrhythmias such as ventricular tachycardia. These arrhythmias can progress to ventricular fibrillation (VF), a fatal rhythm requiring urgent intervention. The most common method for preventing fatal arrhythmias is the use of ICDs for primary or secondary prevention in at-risk patients [6,7]. ICDs detect life-threatening arrhythmias and provide treatment by regulating the rhythm with antitachycardia pacing (ATP) or by delivering an internal shock in the event of VF [6]. ICDs are not only therapeutic devices but also play an important role in monitoring and diagnosing patients by recording electrocardiogram (ECG) data [8,9].

Despite their advanced technological features, the use of ICDs can cause various physical, psychological, and social difficulties in patients, such as fear of shock, fear of pain, fear of death, sleep and lifestyle problems, sexual concerns, driving restrictions, and excessive protectiveness of the spouse [10,11,12,13]. These patients are quite heterogeneous in terms of age, gender, socioeconomic status, and comorbidities [14,15,16,17]. This heterogeneity can influence the type and severity of symptoms experienced as well as determine how individuals cope with symptoms [15,16,17,18]. When clinical presentations are examined, ICD patients exhibit a wide range of cardiac and non-cardiac symptoms [19,20].

Understanding the symptom experiences of individuals with ICD not only facilitates clinical monitoring and treatment planning but also aids in determining targeted interventions and personalized symptom management [21,22]. Therefore, it is important for individuals living with ICD to undergo a comprehensive assessment in both medical and sociocultural contexts. Although there are many studies in the literature on the experiences of patients with ICDs, including quality of life, anxiety, and depression levels [23,24,25,26,27], research that jointly assesses symptom frequency, impact on daily life, and culturally framed coping strategies is limited [28,29,30]. Furthermore, most of these studies have been conducted in Western societies, and the impact of cultural and religious differences on symptom management and well-being has only been partially addressed [22,28,29,30]. In the Turkish context, religious beliefs, family solidarity, and social support emerge as key cultural determinants in shaping the perception of symptoms and coping strategies [31,32]. The contribution of this study to the literature is that it not only identifies culturally rooted coping methods such as prayer and herbal tea among ICD patients, but also re-ports their perceived effectiveness using quantitative ratings. Furthermore, the study directly informed the design of a local RCT on lavender inhalation, providing an important example of how coping behaviors observed in a cultural context can be translated into clinical practice. Therefore, understanding not only symptom management but also the sociocultural and spiritual dimensions of these experiences is an integral part of holistic care [31,32].

This study aims to comprehensively evaluate the symptoms experienced by patients with implantable cardioverter defibrillators (ICDs) in Türkiye, the factors affecting these symptoms, and the non-pharmacological methods they use to cope. The findings emphasize the importance of cultural sensitivity in nursing practice and aim to contribute to holistic care approaches that support patients’ quality of life and psychosocial well-being. Furthermore, the findings from this study have laid the groundwork for a secondary study examining the effect of aromatherapy intervention on the most common symptoms [33].

### Research Questions

Is there a significant change in the presence and/or frequency of symptoms reported by patients before and after ICD implantation?

What individual coping methods do ICD patients use to manage their symptoms?

What is the perceived effectiveness of the coping methods used by ICD patients?

Is there a significant relationship between symptom changes and demographic characteristics in ICD patients?

## 2. Materials and Methods

### 2.1. Type, Location, and Time of the Study

This observational, cross-sectional study was conducted between May 2022 and August 2022 at the arrhythmia outpatient clinic of a university hospital in eastern Türkiye. The target population consisted of ICD patients who visited the arrhythmia outpatient clinic for outpatient treatment during the specified dates. Due to our inability to directly measure patients’ symptoms prior to ICD implantation and the limited availability of clinical data for this period, pre-implantation symptoms were assessed retrospectively after the procedure, while post-implantation symptoms were assessed concurrently. Aware that retrospective reporting may lead to recall bias, we used structured questions to minimize this effect and, where possible, reviewed medical records (e.g., clinical notes, emergency department visits, medication changes) to verify the presence or absence of symptoms. Triangulation was achieved to a limited extent by evaluating participant statements alongside medical records; however, as the level of detail in the records was not the same for all patients, this verification could only be applied to a portion of the sample. Therefore, the possibility of recall bias cannot be completely ruled out. Furthermore, retrospective symptom assessment may reflect both actual change and potential memory impairment or emotional reappraisal; therefore, our findings should be interpreted as changes reported by patients. The study was also retrospectively registered on ClinicalTrials.gov (Identifier: NCT07121855; https://clinicaltrials.gov/study/NCT07121855). The study was conducted in accordance with the STROBE reporting guidelines (Appendix A).

### 2.2. Population and Sample

The target population consists of ICD patients who visited the arrhythmia outpatient clinic between May 2022 and August 2022. The known population sampling method was used to identify eligible participants. Approximately 689 patients monitored with ICDs annually are present at the hospital where the study was conducted. Patients presenting during the 3-month period were considered the population, and the population size was estimated as N ≈ 172, assuming this number represents approximately one-fourth of the annual patient count. The minimum sample size was calculated as 119 patients using the formula for sample size calculation for a known population, assuming a 95% confidence level (*Z* = 1.96), a 5% sampling error (*d* = 0.05), and *p* = 0.05.



$$n=\frac{n.Z^{2}.p.q}{d^{2}.\left(N-1\right)+Z^{2}.p.q}$$



The minimum sample size was calculated as 119 patients. Purposeful sampling was used to achieve the sample size. In this context, ICD patients who visited the arrhythmia outpatient clinic and met the study criteria were identified by the researchers directly during their outpatient visits and invited to participate in the study. Communication with suitable patients was established face-to-face during the outpatient clinic visit; participation was voluntary. A total of 145 ICD patients were reached during the specified period. Three patients who provided incomplete responses were not included in the study. The study was completed with 142 patients (Figure 1).

#### 2.2.1. Inclusion Criteria

Patients who were informed about the study and voluntarily signed the informed consent form,Patients who had an ICD implanted at least three months prior to study enrollment,Patients without communication difficulties (able to speak Turkish and without cognitive impairment),Patients aged 18 years and older [15,22,24,34,35]

#### 2.2.2. Exclusion Criteria

Patients unable to sign the informed consent form,Patients who have undergone ICD implantation within the last three months,Patients with communication difficulties,Patients under 18 years of age,Those with symptoms that cannot be directly attributed to the ICD due to serious physical illnesses such as severe anemia and cancer [15,22,24,34,35].

### 2.3. Data Collection

In this study, data were collected by researchers using a structured data collection form based on a review of the literature [15,22,24,36,37,38]. Data were collected through face-to-face interviews. The form consists of three sections:

#### 2.3.1. Information Obtained from Medical Records

This section contains basic information about the patient’s clinical condition and ICD history: comorbidity status, ICD implantation date, ejection fraction (%), and New York Heart Association (NYHA) classification.

#### 2.3.2. Sociodemographic Information

Questions directed at the patient assessed gender, age, education level, marital status, family type, employment status, occupation, place of residence, perception of access to healthcare services, income status, current smoking status, and regular use of prescribed medications.

#### 2.3.3. Symptom Experiences and Coping Methods

This section evaluates the frequency of symptoms experienced by patients and the methods they use to manage their symptoms. The frequency of symptoms experienced before and after ICD implantation, such as chest pain, palpitations, dizziness, syncope, shortness of breath, fatigue, insomnia, fear, and anxiety, was measured using the codes “none = 0”, “sometimes = 1”, “often = 2”, and “always = 3”. There is no validated symptom severity scale specific to ICD patients in the literature. However, the ICD Patient Concerns Assessment scale evaluates the frequency of 32 symptoms experienced by ICD patients and rates each symptom on a Likert-type scale ranging from 0 (never) to 7 (every day) [39]. In this study, the symptom items were adapted by the researchers based on statements found in this and similar previous scales and structured based on the literature [22,39]. The form was tested on 10 pilot patients before implementation, and the pilot data were not included in the study.

Additionally, the methods patients used to manage their symptoms (nutritional supplements, herbal teas, prayer, walking, exercise, meditation, yoga, breathing exercises, relaxation techniques, massage, aromatherapy, reflexology, music therapy, etc.) were evaluated, and the perceived effectiveness of these methods was assessed using the options “not effective = 0”, “1 = partially effective”, and “2 = effective”. In this study, perceived effectiveness was assessed by researchers using a single-item form (“How effective do you think this method is in managing your symptoms?”). Although the items assessing coping methods in the literature are not specific to the ICD patient population, they were structured by adapting the coping strategies approach used in the Coping Strategies Questionnaire–Revised (CSQ-R). The CSQ-R is a 27-item, 7-point Likert scale instrument that includes strategies such as catastrophizing, self-reassurance, distraction, distancing, sensory avoidance, and prayer [40,41]. In this study, items related to symptom management were adapted from this framework and applied to ICD patients, with the addition of complementary–integrative coping methods. In addition, the perceived effectiveness assessment item was tested in 10 patients in the pilot study, and the pilot data were not included in the analyses.

After the pilot studies, the necessary adjustments were made to the forms and the forms were finalized. The limitations of single-item assessments are discussed in the discussion section, and the use of measurement tools specific to ICD patients and validated for future studies is recommended.

### 2.4. Data Analysis

Statistical analyses were performed using SPSS 26.0 (IMB Corp., Armoni, NY, USA) software at a 95% confidence level. Categorical data were described using frequency and percentage, while continuous numerical data were described using mean and standard deviation. Relationships between categorical variables were evaluated using the chi-square test. Measurements were analyzed using the ANOVA test in terms of coping methods, while changes in symptom intensity before and after ICD implantation were analyzed using the Wilcoxon test because the distribution of reported symptom scores reflected an ordinal data structure. The change in symptom severity before and after implantation was further examined by calculating the mean difference and corresponding 95% confidence intervals. Subgroup analysis was also conducted by classifying participants according to ICD shock status (shocked or non-shocked); pre- and post-implantation symptom scores were reported separately for each subgroup, and mean differences were presented. As the analysis was exploratory and descriptive in nature, no interaction test or formal hypothesis testing was performed between subgroups.

## 3. Results

A total of 142 patients were included in the study. The majority of participants were male (72.5%) and the mean age was 61.96 ± 11.93. The educational level of the vast majority of participants (82.7%) was high school/vocational high school or lower (82.7%). Most patients were married (83.1%) and lived in a nuclear family (85.2%). The rate of unemployed participants was 78.9%. In terms of residence, the majority lived in district centers (59.2%). More than half of the participants reported easy access to health services (52.1%). The most common income level was “middle” (56.3%). The proportion of current smokers was 21.8%. Most participants took their medication regularly (90.1%). The presence of comorbidities was quite common (91.5%). The mean ICD implantation duration was 4.68±5.45 years, and the mean ejection fraction was 32.89 ± 3.49%. According to the New York Heart Association (NYHA) classification, most patients were in class II (57.7%). Regarding the history of ICD shocks, 35.2% of patients reported experiencing at least one shock, while 64.8% reported not experiencing any shocks. Detailed descriptive findings are presented in Table 1.

A total of 142 patients retrospectively reported symptom severity before and after ICD implantation. A significant proportion of patients reported experiencing various cardiac symptoms prior to implantation. For chest pain, 58.5% of patients selected “sometimes, often, or always”; for palpitations, this rate was 76.8%. Similarly, dyspnea was reported as “sometimes” or more severe in 78.9% of patients. Fatigue was also a frequently reported symptom; 90.8% of participants reported experiencing fatigue “sometimes, often, or always” prior to implantation. Although reports of dizziness and syncope were lower, 38.0% and 29.6% of patients, respectively, reported experiencing these symptoms with some frequency prior to implantation. Insomnia (28.9%), fear (31.0%), and anxiety (40.8%) were also among the symptoms reported prior to implantation. After ICD implantation, patients reported some cardiac symptoms less frequently. The “none” response for chest pain increased from 41.5% to 66.9%; for palpitations, this rate rose from 23.2% to 60.6%. A marked decrease was also noted in dizziness and syncope symptoms; “none” reports for dizziness increased from 62.0% to 76.1%, and for syncope from 70.4% to 88.7%. Similarly, milder reports were observed for dyspnea; the “none” rate increased from 21.1% to 42.3%. In contrast, patients reported a retrospective increase in fatigue, with most rating this as “frequent” after implantation (58.5%). No significant change was observed in insomnia (68.3% ‘none’) or fear (72.5% “none”). However, patients reported higher levels of anxiety, with 59.2% describing it as “frequent” after implantation (Table 2).

According to the Wilcoxon test results, when retrospective changes in symptom severity reported after ICD implantation were examined, patients reported significant reductions in various symptoms. Chest pain severity decreased significantly (before: 0.86 ± 0.88; after: 0.36 ± 0.55; mean difference: 0.50 [95% CI: 0.36–0.64]; *p* < 0.001). Similarly, patients reported a significant reduction in palpitations (before: 1.44 ± 1.03; after: 0.42 ± 0.54; mean difference: 1.02 [0.84–1.20]; *p* < 0.001). Dizziness also decreased significantly (before: 0.49 ± 0.72; after: 0.25 ± 0.45; mean difference: 0.25 [0.12–0.37]; *p* < 0.001) and syncope (before: 0.35 ± 0.59; after: 0.12 ± 0.35; mean difference: 0.23 [0.13–0.33]; *p* < 0.001). Shortness of breath severity also decreased similarly according to patient reports (before: 1.42 ± 1.01; after: 0.73 ± 0.74; mean difference: 0.68 [0.51–0.85]; *p* < 0.001). In contrast, patients reported an increase in fatigue severity after ICD implantation (before: 1.59 ± 0.87; after: 1.97 ± 0.66; mean difference: −0.38 [−0.52 to −0.24]; *p* < 0.001). No significant difference was observed in insomnia scores (before: 0.38 ± 0.68; after: 0.42 ± 0.70; mean difference: −0.04 [−0.12 to 0.05]; *p* = 0.473). Similarly, no significant change was observed in fear levels (before: 0.39 ± 0.66; after: 0.31 ± 0.55; mean difference: 0.08 [−0.01 to 0.16]; *p* = 0.082). Unlike most symptoms, patients reported a significant increase in anxiety levels after implantation (before: 0.54 ± 0.74; after: 1.80 ± 0.61; mean difference: −1.27 [−1.39 to −1.14]; *p* < 0.001) (Table 3, Figure 2).

According to the Wilcoxon test results, when retrospective changes in symptom severity reported after ICD implantation were examined according to shock status, patients who experienced shock reported significant reductions in various symptoms after implantation. Chest pain intensity decreased (pre: 1.04 ± 0.92; post: 0.40 ± 0.64; mean difference: 0.64 [95% CI: 0.40–0.88]; *p* < 0.001), palpitations also decreased (pre: 1.62 ± 1.03; post: 0.52 ± 0.58; mean difference: 1.10 [0.75–1.45]; *p* < 0.001), dizziness (pre: 0.44 ± 0.67; post: 0.18 ± 0.39; mean difference: 0.26 [0.05–0.47]; *p* = 0.014), and shortness of breath (before: 1.54 ± 1.03; after: 0.80 ± 0.78; mean difference: 0.74 [0.47–1.01]; *p* < 0.001). No statistically significant change was observed in syncope (*p* = 0.201). However, patients reported fatigue (before: 1.84 ± 0.87; after: 2.10 ± 0.71; mean difference: −0.26 [−0.50 to −0.02]; *p* = 0.047) and anxiety (before: 0.74 ± 0.88; after: 1.80 ± 0.64; mean difference: −1.06 [−1.30 to −0.82]; *p* < 0.001). No significant changes were reported in insomnia (*p* = 0.405) or fear (*p* = 0.317). Similar patterns were observed in symptom changes reported by patients who did not experience shock. Chest pain decreased significantly (before: 0.76 ± 0.84; after: 0.34 ± 0.50; mean difference: 0.42 [0.25–0.60]; *p* < 0.001), palpitations (before: 1.34 ± 1.03; after: 0.36 ± 0.50; mean difference: 0.98 [0.78–1.18]; *p* < 0.001), dizziness (before: 0.52 ± 0.75; after: 0.28 ± 0.48; mean difference: 0.24 [0.08–0.40]; *p* = 0.004), syncope (before: 0.36 ± 0.60; after: 0.07 ± 0.25; mean difference: 0.29 [0.18–0.41]; *p* < 0.001) and shortness of breath (before: 1.35 ± 0.99; after: 0.70 ± 0.72; mean difference: 0.65 [0.43–0.87]; *p* < 0.001). In contrast, patients reported increased fatigue (before: 1.46 ± 0.84; after: 1.90 ± 0.63; mean difference: −0.45 [−0.62 to −0.27]; *p* < 0.001) and increased anxiety (before: 0.42 ± 0.63; after: 1.80 ± 0.60; mean difference: −1.38 [−1.52 to −1.24]; *p* < 0.001). Insomnia (*p* = 0.108) and fear (*p* = 0.157) did not show significant changes in this group (Table 4).

Table 5 shows the distribution of coping strategies used and reported by the patients. Within the religious/spiritual coping category, 95 patients (66.9%) reported using the prayer/remembrance method. Among these, 70 patients (73.7%) stated that this method was effective, 20 patients (21.1%) stated that it was partially effective, and 5 patients (5.2%) stated that it was not effective. Within the physical activity category, walking was used by 34 patients (23.9%). Of those who used this method, 25 (73.5%) rated it as effective, 7 (20.6%) rated it as somewhat effective, and 2 (5.9%) rated it as ineffective. In the complementary/traditional methods category, 13 patients (9.2%) reported drinking herbal tea. Among these participants, 5 (38.5%) found it effective, 8 (61.5%) found it partially effective, and none found it ineffective.

The chi-square test showed a significant relationship between marital status and coping methods (*p* < 0.05). 65.3% of married individuals used prayer, 9.3% used herbal tea, and 25.4% used walking as coping methods. Among single individuals, 25% preferred prayer, 0% preferred herbal tea, and 75% preferred walking. Among widowed individuals, the rate of praying was 85%, the rate of drinking herbal tea was 10%, and the rate of walking was 5%. Furthermore, the chi-square test revealed no statistically significant difference between coping methods and variables such as gender, education level, family type, employment status, occupation, place of residence, access to healthcare, income status, smoking status, regular medication use, comorbidity status, NYHA classification, shock status, and age (*p* > 0.05). The ANOVA test showed no significant difference between patients who used prayer, herbal tea consumption, and walking as coping methods in terms of ICD implantation duration and ejection fraction (Table 6).

## 4. Discussion

Implantable cardioverter defibrillators (ICDs) are widely used due to their clinical effectiveness in preventing life-threatening arrhythmias; however, it is known that patient experiences of living with an ICD are still limited in the literature. This gap particularly points to an important information void in terms of symptom management and psychosocial adjustment. This study is one of the first to quantitatively describe symptom trajectories and culturally shaped coping strategies in individuals with ICDs in Türkiye.

In our study, patients reported significant reductions in chest pain, palpitations, dizziness, syncope, and shortness of breath in their retrospectively reported symptoms after ICD implantation (*p* < 0.001). These results support the effect of ICD in alleviating the burden of cardiac symptoms, as demonstrated in previous studies [42,43]. The fact that most patients, both those who experienced shocks and those who did not, reported a similar reduction in cardiac symptoms after implantation reinforces this trend. However, a significant increase in fatigue and anxiety levels was observed after ICD implantation, both in the general sample and according to shock status (*p* < 0.001). The increase in fatigue and anxiety after implantation, even as cardiac symptoms improved, is a noteworthy finding. This situation demonstrates that the symptom burden includes not only a physical but also a significant psychological component. Indeed, the literature reports that energy and fatigue levels in individuals using permanent cardiac devices mostly range from moderate to low [24,34,44,45], which appears consistent with the increase in fatigue observed in our study.

It is known that fatigue is a common symptom that negatively affects quality of life in patients with chronic heart disease [17]. In addition to psychological adjustment difficulties, the multiple systemic effects of cardiovascular disease form the physiological basis of fatigue, limiting daily activities by reducing physical capacity in patients; functionality declines even more markedly when accompanied by shortness of breath and exercise intolerance [24,46]. In ICD patients, various psychosocial and physiological factors such as post-device conditioning loss, activity restrictions, shock anticipation, excessive focus on somatic sensations, protective attitudes of family members, and role changes may contribute to this picture [12,13,21,24,34,37,44,45,47]. Increased fatigue is therefore a multidimensional symptom arising from the simultaneous effects of both physiological and psychosocial mechanisms.

It has been reported that the most prominent problem area in ICD patients is psychological effects [48,49]. In this context, one study reported that depression and anxiety are common in ICD patients and that this condition can negatively affect clinical outcomes by exacerbating heart failure, increasing the risk of arrhythmia, and, in particular, that depression can worsen the frequency of arrhythmic events and mortality [25]. Indeed, the current literature also shows that ICD patients generally experience moderate to high levels of anxiety and depression [10,11,26,50]. These results point to the seriousness of mental health problems in the ICD patient population. The increase in anxiety after implantation in our study also supports this trend in the literature. Our findings may be related to lifestyle restrictions that may arise after ICD implantation, the adaptation process associated with the physical presence of the device, perceived social support, concerns about work and social life, and potential psychosocial factors [13,21,51,52,53].

When symptom progression was analyzed according to shock experience in the study, most patients who received shocks and those who did not reported similar reductions in most cardiac symptoms after implantation; this suggests that ICD treatment contributes to symptom stabilization regardless of shock exposure. Furthermore, contrary to findings reported by Ghezzi et al. (2023) [10] showing significantly higher levels of anxiety and depression in ICD patients who experienced shocks, our study found no significant difference in anxiety severity between patients who experienced shocks and those who did not. This discrepancy may be related to our retrospective within-patient design, the relatively low rate of shock exposure in our sample, and potential moderators related to cultural or coping styles that may mitigate the emotional impact of shocks. Therefore, while the collective evidence suggests that shocks increase psychological distress, our findings indicate that this relationship may not be universal across all patient populations.

This study, in addition to existing syntheses on symptom experiences of ICD patients [10,22], also evaluated the individual strategies patients used to cope with their symptom experiences. When examining individual coping strategies, it was found that patients most frequently resorted to religious/spiritual methods (especially prayer), followed by physical activity (walking) and complementary/traditional practices (consumption of herbal tea). A meta-synthesis of qualitative studies reported that ICD patients developed both passive (avoidance, suppression) and active (acceptance, reinterpretation, spiritual empowerment) coping strategies [30]. The high use of prayer in the Turkish cultural context points to a culturally embedded mechanism that strengthens patients’ subjective sense of control and facilitates coping with uncertainty [54,55]. However, evidence regarding the contribution of coping strategies to long-term adaptation is limited.

The finding in our study that marital status significantly affected coping strategies suggests that social roles and family structure may play a decisive role in symptom management. This result is consistent with the literature emphasizing the relationship between social support and spiritual coping with psychological factors and coping [56,57]. Furthermore, our study found no significant relationship between coping strategies and sociodemographic or clinical variables such as gender, education, comorbidities, or cardiac function. This indicates that religious coping is a widely adopted cultural mechanism among different Turkish patient profiles.

From a nursing practice perspective, these findings highlight the importance of psychosocial assessment, culturally sensitive interventions, and patient education in post-ICD care. Nurses play a critical role in identifying patients’ emotional distress, supporting effective coping strategies, and integrating spiritual needs into the care plan.

In conclusion, this study provides valuable insights into symptom progression and coping behaviors among Turkish ICD patients. The co-occurrence of physiological recovery and psychological distress underscores the importance of holistic and culturally focused nursing approaches that address both the medical and existential dimensions of living with an ICD.

## 5. Future Directions

This study demonstrates that understanding symptom experience and coping patterns in ICD patients is critical to the development of holistic care approaches. An aromatherapy RCT (NCT06874777) conducted and completed based on these findings demonstrated that lavender inhalation provided clinically meaningful benefits. Two drops of lavender inhalation were administered to ICD patients for two minutes before bedtime for one month; a total of five assessments were conducted, including baseline assessment and weekly follow-ups. The intervention group showed a more pronounced decrease in fatigue and anxiety levels during follow-ups in the 2nd, 3rd, and 4th weeks (*p* < 0.05), indicating that aromatherapy has the potential to be a feasible and effective supportive intervention for ICD patients [33].

## 6. Limitations

Several limitations should be considered when interpreting the findings of this study. First, data on symptoms and coping strategies were collected based on self-reports through medical records, questionnaires, and interviews, which may lead to recall bias. Retrospective assessment of pre-implantation symptoms carries the risk of recall bias. To mitigate this effect, patients’ statements were cross-checked with medical records to a limited extent whenever possible; however, since medical records were not available at the same level of detail for all participants, this verification could only be performed for a portion of the sample. Therefore, the direction and possible magnitude of recall bias can only be assessed with partial reliability. Furthermore, retrospective symptom assessment may reflect memory bias in addition to actual change; therefore, findings should be interpreted only as reported changes. Second, the cross-sectional design, which assessed symptoms and coping strategies at a single point in time, precludes establishing causal relationships between variables. Third, the study did not use validated scales for symptom assessment items and perceived efficacy measurement; instead, researchers adapted tools from the literature. Although these tools were tested in a pilot study prior to implementation, single-item assessments and adapted scales have psychometric limitations. Fourth, these findings may not be generalizable to countries or cultural contexts with differing religious or spiritual profiles or to populations living within different healthcare system structures. Sixth, the study was conducted on a single-center sample in Turkey, and cross-cultural differences may influence symptom perception, spiritual coping methods, and the use of complementary methods, which should be considered. Finally, objective clinical measurements (e.g., arrhythmia burden via device interrogation) were not used to validate self-reported symptoms in the study. To understand symptom patterns and coping mechanisms in ICD patients more comprehensively, larger, multicenter, and longitudinal studies incorporating both subjective and objective measurements are required.

## 7. Conclusions

In this cross-sectional study, we evaluated changes in symptoms reported retrospectively by patients before and after ICD implantation. Our findings showed a significant decrease in cardiac symptoms such as chest pain, palpitations, dizziness, syncope, and shortness of breath; however, fatigue and anxiety increased significantly, while no significant change was observed in insomnia or fear. Furthermore, changes in symptom burden according to shock status were consistent with these findings. These findings suggest that ICDs are effective in controlling cardiac symptoms but may also impose a psychological burden. Therefore, psychological assessment and support are as important as cardiac monitoring in ICD patients. Furthermore, our findings emphasize the necessity of holistic, patient-centered approaches that encompass psychological, cultural, and spiritual needs. From a nursing perspective, it is critically important to recognize patients’ emotional distress early, support effective coping methods, and integrate patients’ belief systems into the care plan. In this context, complementary and integrative practices such as mind–body interventions or relaxation-based therapies should be considered as supportive approaches to enhance psychological well-being, alongside traditional biomedical monitoring and spiritually based coping methods.

## Figures and Tables

**Figure 1 healthcare-14-00059-f001:**
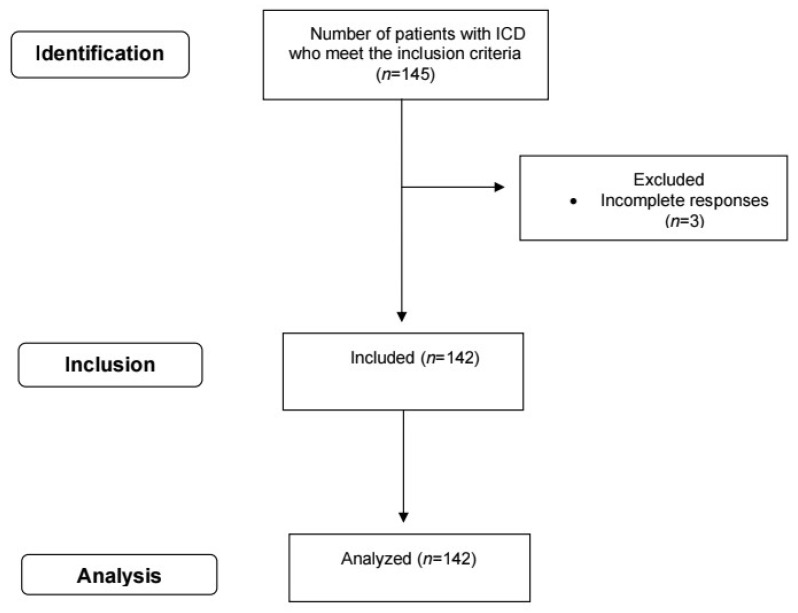
Flowchart.

**Figure 2 healthcare-14-00059-f002:**
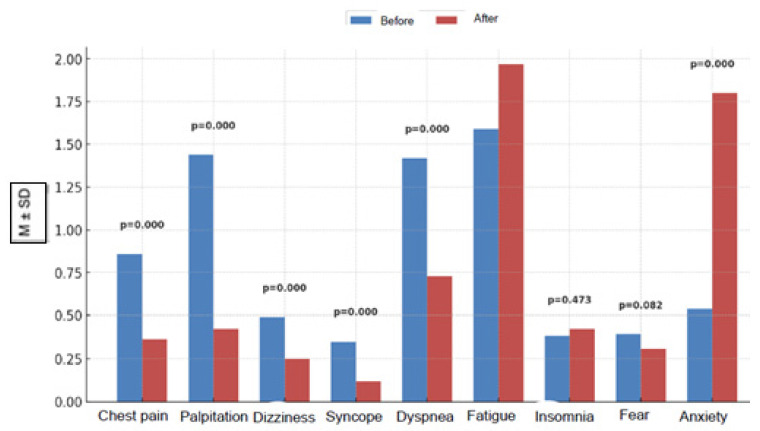
Retrospectively self-reported mean symptom scores before and after implantation (with *p*-values).

**Table 1 healthcare-14-00059-t001:** Descriptive characteristics of participants (*n* = 142).

Characteristics		*n* (%)
Biological sex	Male	103 (72.5)
	Female	39 (27.5)
Educational level	Illiterate	23 (16.2)
	Literate	5 (3.5)
	Elementary school graduate	74 (52.1)
	Middle school graduate	14 (9.9)
	High school graduate	19 (13.4)
	University graduate	6 (4.2)
	Master’s/Doctorate	1 (0.7)
Marital status	Married	118 (83.1)
	Single	4 (2.8)
	Widowed	20 (14.1)
Family type	Nuclear family	121 (85.2)
	Extended family	18 (12.7)
	Single-person household	3 (2.1)
Employment status	Employed	21 (14.8)
	Unemployed	121 (85.2)
Occupation	Civil servant	3 (2.1%)
	Worker	15 (10.6%)
	Self-employed	12 (8.5%)
	Retired	90 (63.4%)
	Housewife	22 (15.5%)
Place of residence	Metropolitan	18 (12.7)
	Province	19 (13.4)
	District	84 (59.2)
	Town	2 (1.4)
	Village	19 (13.4)
Access to healthcare services	Easy	74 (52.1)
	Difficult	68 (47.9)
Income status	Very good	0 (0)
	Good	11 (7.7)
	Average	80 (56.3)
	Poor	39 (27.5)
	Very poor	12 (8.5)
Smoking status	Smoker	31 (21.8)
	Non-smoker	42 (29.6)
	Ex-smoker	69 (48.6)
Regular medication use status	User	128 (90.1)
	Partial user	10 (7)
	Non-user	4 (2.8)
Comorbidity status	Present	130 (91.5)
	None	12 (8.5)
NYHA classification	Class II	82 (57.7)
	Class III	59 (41.5)
	Class IV	1 (0.7)
Shock status	Shocked	50 (35.2%)
	Non-shocked	92(64.8%)
		$\bar{x}$ *± SD*
Age		61.96 ± 11.93
ICD Implantation Duration		4.68 ± 5.45
Ejection fraction		32.89 ± 3.49

*n*, Number; *%*, Percentage; *SD*, Standard deviation; $\bar{x}$, Mean.

**Table 2 healthcare-14-00059-t002:** Retrospective distribution of self-reported symptom severity before and after ICD implantation (*n* = 142).

Symptoms		None	Sometimes	Frequently	Always
		*n (%)*	*n (%)*	*n (%)*	*n (%)*
Chest pain	Pre-implantation	59 (41.5)	51 (35.9)	25 (17.6)	7 (4.9)
	Post-implantation	95 (66.9)	44 (31)	2 (1.4%)	1 (0.7)
Palpitations	Pre-implantation	33 (23.2)	39 (27.5)	45 (31.7%)	25 (17.6)
	Post-implantation	86 (60.6)	53 (37.3)	3 (2.1%)	0 (0)
Dizziness	Pre-implantation	88 (62%)	41 (28.9)	10 (7%)	3 (2.1)
	Post-implantation	108 (76.1)	33 (23.2)	1 (0.7%)	0
Syncope	Pre-implantation	100 (70.4%)	34 (23.9)	8 (5.6)	0 (0%)
	Post-implantation	126 (88.7%)	15 (10.6)	1 (0.7%)	0 (0)
Dyspnea	Pre-implantation	30 (21.1%)	47 (33.1)	41 (28.9)	24 (16.9)
	Post-implantation	60 (42.3)	63 (44.4)	16 (11.3)	3 (2.1)
Fatigue	Pre-implantation	13 (9.2%)	55 (38.7%)	51 (35.9%)	23 (16.2)
	Post-implantation	1 (0.7%)	30 (21.1%)	83 (58.5%)	28 (19.7)
Insomnia	Pre-implantation	101 (71.1%)	31 (21.8)	7 (4.9%)	3 (2.1)
	Post-implantation	97 (68.3)	34 (23.9)	8 (5.6%)	3 (2.1)
Fear	Pre-implantation	98 (69%)	36 (25.4)	5 (3.5%)	3 (2.1)
	Post-implantation	103 (72.5)	35 (24.6)	3 (2.1%)	1 (0.7)
Anxiety	Pre-implantation	84 (59.2%)	43 (30.3%)	12 (8.5%)	3 (2.1)
	Post-implantation	0 (0%)	43 (30.3%)	84 (59.2%)	15 (10.6)

*n*, Number; %, Percentage.

**Table 3 healthcare-14-00059-t003:** Retrospectively self-reported changes in symptom severity before and after ICD implantation (*n* = 142).

Symptoms	Pre-Implantation $\bar{x}$ *± SD*	Post-Implantation $\bar{x}$ *± SD*	Mean Difference %95 *CI*	*p*
Chest pain	0.86 ± 0.88	0.36 ± 0.55	0.50 (0.36; 0.64)	<0.001
Palpitations	1.44 ± 1.03	0.42 ± 0.54	1.02 (0.84; 1.20)	<0.001
Dizziness	0.49 ± 0.72	0.25 ± 0.45	0.25 (0.12; 0.37)	<0.001
Syncope	0.35 ± 0.59	0.12 ± 0.35	0.23 (0.13; 0.33)	<0.001
Dyspnea	1.42 ± 1.01	0.73 ± 0.74	0.68 (0.51; 0.85)	<0.001
Fatigue	1.59 ± 0.87	1.97 ± 0.66	−0.38 (−0.52; −0.24)	<0.001
Insomnia	0.38 ± 0.68	0.42 ± 0.7	−0.04 (−0.12; 0.05)	0.473
Fear	0.39 ± 0.66	0.31 ± 0.55	0.08 (−0.01; 0.16)	0.082
Anxiety	0.54 ± 0.74	1.8 ± 0.61	−1.27 (−1.39; −1.14)	<0.001

*SD*, Standard deviation; $\bar{x}$, Mean; *CI*, Confidence interval. Wilcoxon test.

**Table 4 healthcare-14-00059-t004:** Comparison of self-reported symptom severity before and after implantation among participants according to shock status (*n* = 142).

Symptoms		Shock Status	
		Shocked	Non-Shocked
		$\bar{x}$ *± SD*	$\bar{x}$ *± SD*
Chest pain	Pre-implantation	1.04 ± 0.92	0.76 ± 0.84
	Post-implantation	0.4 ± 0.64	0.34 ± 0.5
	Mean Difference (%95 *CI*)	0.64 (0.40; 0.88)	0.42 (0.25; 0.60)
	*p*	<0.001	<0.001
Palpitations	Pre-implantation	1.62 ± 1.03	1.34 ± 1.03
	Post-implantation	0.52 ± 0.58	0.36 ± 0.5
	Mean Difference (%95 *CI*)	1.10 (0.75; 1.45)	0.98 (0.78; 1.18)
	*p*	<0.001	<0.001
Dizziness	Pre-implantation	0.44 ± 0.67	00.52 ± 0.75
	Post-implantation	0.18 ± 0.39	0.28 ± 0.48
	Mean Difference (%95 *CI*)	0.26 (0.05; 0.47)	0.24 (0.08; 0.40)
	*p*	0.014	0.004
Syncope	Pre-implantation	0.34 ± 0.56	0.36 ± 0.6
	Post-implantation	0.22 ± 0.46	0.07 ± 0.25
	Mean Difference (%95 *CI*)	0.12 (−0.07; 0.31)	0.29 (0.18; 0.41)
	*p*	0.201	<0.001
Dyspnea	Pre-implantation	1.54 ± 1.03	1.35 ± 0.99
	Post-implantation	0.8 ± 0.78	0.7 ± 0.72
	Mean Difference (%95 *CI*)	0.74 (0.47; 1.01)	0.65 (0.43; 0.87)
	*p*	<0.001	<0.001
Fatigue	Pre-implantation	1.84 ± 0.87	1.46 ± 0.84
	Post-implantation	2.1 ± 0.71	1.9 ± 0.63
	Mean Difference (%95 *CI*)	−0.26 (−0.50; −0.02)	−0.45 (−0.62; −0.27)
	*p*	0.047	<0.001
Insomnia	Pre-implantation	0.42 ± 0.64	0.36 ± 0.7
	Post-implantation	0.36 ± 0.63	0.45 ± 0.73
	Mean Difference (%95 *CI*)	0.06 (−0.09; 0.21)	−0.09 (−0.19; 0.02)
	*p*	0.405	0.108
Fear	Pre-implantation	0.52 ± 0.76	0.32 ± 0.59
	Post-implantation	0.44 ± 0.64	0.24 ± 0.48
	Mean Difference (%95 *CI*)	0.08 (−0.07; 0.23)	0.08 (−0.03; 0.18)
	*p*	0.317	0.157
Anxiety	Pre-implantation	0.74 ± 0.88	0.42 ± 0.63
	Post-implantation	1.8 ± 0.64	1.8 ± 0.6
	Mean Difference (%95 *CI*)	−1.06 (−1.30; −0.82)	−1.38 (−1.52; −1.24)
	*p*	<0.001	<0.001

*SD*, Standard deviation; $\bar{x}$, Mean; *CI*, Confidence interval. Wilcoxon test.

**Table 5 healthcare-14-00059-t005:** Distribution of individual coping methods self-reported by participants (*n* = 142).

Coping Category	Coping Method	Number of Uses *n* (%)	Effectiveness *n* (%)
Religious/Spiritual	Prayer/Remembrance	95 (66.9)	Effective: 70 (73.7) Partially: 20 (21.1) Not effective: 5 (5.2)
Physical Activity	Walking	34 (23.9)	Effective: 25 (73.5) Partially: 7 (20.6) Not effective: 2 (5.9)
Complementary/Traditional	Drinking herbal tea	13 (9.2)	Effective: 5 (38.5) Partially: 8 (61.5) Not effective: 0 (0)

*n*, Number; %, Percentage.

**Table 6 healthcare-14-00059-t006:** Comparison of descriptive and clinical characteristics according to individual coping methods (*n* = 142).

Characteristics	Prayer/Zikr *n* (%)	Herbal Tea Consumption *n* (%)	Walking *n* (%)	*p*
Biological sex				0.062 *
Male	63 (61.2)	11 (10.7)	29 (28.2)	
Female	32 (82.1)	2 (5.1)	5 (12.8)	
Education Level				0.554 *
Illiterate	17 (73.9)	2 (8.7)	4 (17.4)	
Literate	5 (100)	0 (0)	0 (0)	
Elementary school graduate	52 (70.3)	6 (8.1)	16 (21.6)	
Middle school graduate	8 (57.1)	2 (14.3)	4 (28.6)	
High school graduate	8 (42.1)	3 (15.8)	8 (42.1)	
University graduate	4 (66.7)	0 (0)	2 (33.3)	
Master’s/Doctorate	1 (100)	0 (0)	0 (0)	
Marital Status				0.042 *
Married	77 (65.3)	11 (9.3)	30 (25.4)	
Single	1 (25)	0 (0)	3 (75)	
Widowed	17 (85)	2 (10)	1 (5)	
Family Type				0.472 *
Nuclear family	78 (64.5)	11 (9.1)	32 (26.4)	
Extended family	14 (77.8)	2 (11.1)	2 (11.1)	
Living alone	3 (100)	0 (0)	0 (0)	
Employment Status				0.854 *
Working	13 (61.9)	2 (9.5)	6 (28.6)	
Not working	82 (67.8)	11 (9.1)	28 (23.1)	
Occupation				
Civil servant	0	0 (0%)	3 (100%)	0.083 *
Worker	10 (66.7%)	1 (6.7%)	4 (26.7%)	
Self-employed	8 (66.7)	1 (8.3%)	3 (25%)	
Retired	58 (64.4)	10 (11.1%)	22 (24.4%)	
Housewife	19 (86.4)	1 (4.5%)	2 (9.1%)	
Place of Residence				0.909 *
Metropolitan	10 (55.6)	3 (16.7)	5 (27.8)	
Province	14 (73.7)	1 (5.3)	4 (21.1)	
District	57 (67.9)	8 (9.5)	19 (22.6)	
Town	1 (50)	0 (0)	1 (50)	
Village	13 (68.4)	1 (5.3)	5 (26.3)	
Access to Health Services				0.823 *
Easy	49 (66.2)	6 (8.1)	19 (25.7)	
Difficult	46 (67.6)	7 (10.3)	15 (22.1)	
Income Status				0.619 *
Good	6 (54.5)	1 (9.1)	4 (36.4)	
Average	56 (70)	7 (8.8)	17 (21.3)	
Poor	24 (61.5)	3 (7.7)	12 (30.8)	
Very weak	9 (75)	2 (16.7)	1 (8.3)	
Smoking Status				0.523 *
Smoker	23 (74.2)	1 (3.2)	7 (22.6)	
Non-smoker	30 (71.4)	4 (9.5)	8 (19)	
Ex-smoker	42 (60.9)	8 (11.6)	19 (27.5)	
Regular Medication Use				0.090
User	85 (66.4)	10 (7.8)	33 (25.8)	
Partially user	6 (60)	3 (30)	1 (10)	
Not a user	4 (100)	0 (0)	0 (0)	
Comorbidity Status				0.513 *
Current	86 (66.2)	13 (10)	31 (23.8)	
NHYA Classification				0.183 *
Class II	52 (63.4)	5 (6.1)	25 (30.5)	
Class III Shock status Shocked Non-shocked	42 (71.2) 37 (74) 58 (63)	8 (13.6) 3 (6) 10 (10.9)	9 (15.3) 10 (20) 24 (26.1)	0.383 *
Age				
60 years and under	42 (67.7)	4 (6.5)	16 (25.8)	0.592 *
Over 60	53 (66.3)	9 (11.3)	18 (22.5)	
	$\bar{x}$ *± SD*	$\bar{x}$ *± SD*	$\bar{x}$ ± *SD*	
ICD Implantation Duration	4.95 ± 6.28	4.77 ± 3.65	3.91 ± 3.01	0.638 **
Ejection Fraction	32.45 ± 4.74	33.08 ± 3.25	33.24 ± 3.23	0.627 **

*n*, Number; %, Percentage; *SD*, Standard deviation; $\bar{x}$, Mean. * Chi-square test. ** ANOVA test.

## Data Availability

The data used in this study are not publicly available due to privacy principles and ethical committee approval. However, access may be provided upon reasonable request to the relevant author.

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
