# Peer review of "Healthcare2026, 14(1), 59;https://doi.org/10.3390/healthcare14010059"

_healthcare, 2025, doi:10.3390/healthcare14010059_

Round 1

Reviewer 1 Report

Comments and Suggestions for Authors

This review addresses an important, humane question: how patients living with ICDs experience symptoms and use culturally informed coping strategies (notably prayer) in Türkiye. That combination (symptom trajectory plus self-reported coping methods in a non-Western setting) is timely and useful. The study design (cross-sectional, retrospective pre/post symptom recall) is appropriate for an initial, descriptive exploration and the sample (n=142) is decent for a single-center effort.

Paper’s strongest contribution is contextual: it provides evidence that biomedical symptom relief after ICD implantation (chest pain, palpitations, dyspnea) can coexist with rising psychological burden (fatigue and anxiety); and it documents culturally prevalent coping mechanisms (prayer, herbal tea, walking) with basic effectiveness ratings. That is exactly the kind of evidence nursing and holistic-care teams need to design culturally sensitive interventions.

However, the manuscript needs work to be systematic, transparent, and convincing to a broad international readership. Right now the discussion sometimes re-states results as conclusions without rigorous linking back to measurement validity, confounding, or alternative explanations. The methods and results sections would benefit from clearer operational definitions, stronger presentation of effect sizes/uncertainty, and a tighter argument for causality limits imposed by retrospective recall. Language and ordering are mostly understandable, but the manuscript repeats some ideas (particularly the fatigue/anxiety point) in adjacent paragraphs — this weakens the narrative flow and inflates perceived novelty. The handling of statistics is serviceable (Wilcoxon, chi-square, ANOVA), but the reader needs more detail (confidence intervals, multiple comparisons, covariate adjustment) to judge robustness. Finally, the link to the aromatherapy follow-up trial is promising (and a potential impact point) but reads as an afterthought; expand its rationale and methods in a short dedicated paragraph or move some supporting references into Methods/Discussion.

Overall: the manuscript is novel in its cultural focus and local data, but not revolutionary in concept — similar syntheses exist (systematic reviews and meta-analyses on mood, coping, and quality of life in ICD recipients). The current work can add meaningful, practice-relevant detail for Türkiye and similar contexts, if the authors tighten methods, clarify limitations, and reorganize repetitive text.

  1. Novelty & literature placement (important!) 

Authors assert the study fills a gap that most studies are Western (in the Introduction section). The paper does cite major syntheses (e.g., “Perceptions and Experiences…” and a large systematic meta-analysis). Ooi et al. 2016 and Ghezzi et al. 2023 are listed already.

Suggestion: explicitly state how your dataset adds to those reviews (unique cultural measures; prevalence of prayer as coping; planned local RCT on lavender aromatherapy). The current lines where you claim novelty (Introduction, lines ~66–82) are OK but should be more explicit: e.g., “Unlike X and Y, this study… measures culturally framed coping (prayer) and reports effectiveness ratings; it also served as a basis for a randomized aromatherapy trial.” Quote in manuscript: “Furthermore, the findings from this study have laid the groundwork for a secondary study examining the effect of aromatherapy intervention on the most common symptoms…”

2. Design & retrospective symptom recall (must clarify..).

The Methods state: “pre-implant symptoms were evaluated retrospectively after implantation” (lines ~95–99). As quoted on manuscript: “Due to the inability to directly measure patients’ symptoms prior to ICD implantation … pre-implant symptoms were evaluated retrospectively after implantation.”

This introduces recall bias that can drive the observed post-implant increases in anxiety/fatigue (patients may reinterpret pre-implant symptoms differently after diagnosis). The authors mention recall bias in Limitations (lines ~319–327) but should quantify the likely direction and consider sensitivity analysis or triangulation (e.g., check records where available). See Supplement STROBE form: bias item is marked N/A — this is incorrect; address how you assessed or mitigated recall bias.

3. Statistics & effect sizes (improve)

Results report means and p-values (e.g., Table 3: anxiety pre 0.54±0.74 → post 1.8±0.61; p<0.001). Good; but no confidence intervals and no adjustments for multiple testing are provided. Provide 95% CIs for mean changes and consider a false discovery control or explicitly state familywise error approach.

4. Repeated ideas / reordering suggestions (language & flow).
The Discussion repeats the same point about fatigue twice in nearby paragraphs (see lines ~252–256 and ~256–256). Consider consolidating the physiological explanation paragraph with the psychological paragraph that follows. Quited: “However, a significant increase in fatigue severity after implantation (p<0.001) indicates that the symptom burden persists.” and immediately after similar wording appears again.  

Suggestion: Move the paragraph about the aromatherapy RCT from the end of the Discussion into a short dedicated subsection (either Methods if planned/registered or an explicit Future Directions paragraph) so it reads as a prespecified follow-up rather than tacked on. The trial registration NCT07121855 appears on ClinicalTrials.gov; link that and give brief methods. ( https://clinicaltrials.gov/study/NCT07121855 )

  1. Cultural coping & measurement validity (can be improved).

The large majority used prayer (95/142; Table 4) and judged it effective (73.7% effective). As quoted: “Prayer / Remembrance 95 (66.9) Effective: 70 (73.7).”

Question: How was “effective” defined and measured? Was there a validated scale for perceived effectiveness, or a bespoke single item? Please report the item wording and psychometric rationale (if absent, include it in Methods or Supplement). If this is a single subjective item, state limitations and consider possibly adding a short validated scale in future work.

  1. Biomolecular/physiological detail (note gaps)

The manuscript correctly notes lack of objective device data (arrhythmia burden, shock events) as a limitation (lines ~327–329). Quoted:  “Finally, the study did not include objective clinical measurements (e.g., arrhythmia burden obtained from device interrogation) to validate self-reported symptoms.”
Suggestion: either obtain device interrogation data (even aggregated counts of shocks/ATP) or explicitly report whether patients had recent ICD therapies; this helps separate device-related anxiety (shock experience) from generalized anxiety or fatigue.

  1. Figures & tables clarity (minor fix)

Table 2 and Table 3 present useful data but would benefit from clearer labeling (units for means, explanation of score range used for symptoms—what does 0–3 scale mean?). In Table 3 add 95% CI columns. Quote Table 3 lines showing scores.

  1. References & scholarship check

Authors already cited the major systematic reviews/meta-analyses and qualitative syntheses (e.g., Ghezzi 2023 https://doi.org/10.1093/europace/euad130  ; Ooi et al. 2016; Flemme et al. 2012).

there are numerous related syntheses (Ghezzi 2023 Europace; Ooi 2016 https://doi.org/10.1186/s12955-016-0561-0 ; Flemme 2012 ), so the manuscript’s novelty is contextual (Turkish cohort + cultural coping + aromatherapy RCT link), not a wholesale conceptual revolution.

9. Ethics & CONSORT/STROBE reporting.

Supplement includes a STROBE checklist (good) but several items are marked N/A (e.g., bias, missing data handling). Either complete the checklist or justify N/A choices.  

10. Questions:

- For the retrospective pre-implant symptom assessment: can you provide any validation that patients’ recall correlates with medical records for at least a subset (e.g., clinic notes, ER visits, meds)? If not, can you quantify the likely direction and magnitude of recall bias?

-How were “effective/partially/not effective” responses to coping methods operationalized? Please provide the exact questionnaire wording and any prior validation or pilot testing. If no validation exists, consider adding a short paragraph in Methods about face validity and limitations.

-Given the large increase in reported anxiety post-implantation, did you capture whether participants experienced shocks or ATP therapies (device therapies)? If so, present subgroup analysis; if not, please discuss how missing device-therapy data could confound the anxiety/fatigue findings. 

The manuscript is publishable after moderate revisions that address methods transparency (recall bias, instrument wording), statistical presentation (CIs, adjustment for confounders or explicit explanations why not), reorganization to reduce repetition, and a clearer statement of how this work adds to existing syntheses (explicit comparisons to Ghezzi 2023 and other reviews). If the authors implement the specific edits above and answer the l questions, the manuscript should be suitable for acceptance. Bravo!

Author Response

RESPONSE TO REVİEWER 1

  1. Overall assessment – systematicity, transparency, and presentation

Comment: “Manuscript needs to be more systematic, transparent, convincing; discussion repeats results; methods/results require clearer definitions, effect sizes, uncertainty; argument for limits of retrospective recall must be strengthened, language has repetition.”

Response: Thank you for your suggested changes. We have made significant revisions to strengthen the overall flow of the text, reduce repetition, and increase methodological transparency, highlighting our changes in yellow.

Manuscript changes:

  • Repeated paragraphs on fatigue/anxiety in the discussion section were merged and a more logical sequence was created.
  • Operational definitions were clearly added to the methods and results sections (e.g., symptom score ranges, measurement of coping items).
  • Effect sizes (95% CI) and uncertainty measures were added to the table and text.
  • Recall bias and causality limitations were expanded in detail.

  1. Novelty & literature placement

Comment: “Explicitly state how the dataset adds to existing reviews (cultural coping, prayer, aromatherapy RCT link). Lines 66–82 need clearer novelty statement.”

Response: Thank you for your suggested changes. In line with this suggestion, we defined the original value of the study more clearly. Our relationship with existing meta-analyses and systematic reviews was clarified and highlighted in yellow.

Manuscript changes:

The following content was added to the Introduction section:

  • Quantitatively measuring culturally framed coping methods (e.g., prayer) in the Turkish context,
  • Providing an effectiveness assessment,
  • Establishing the findings as a basis for the aromatherapy randomized controlled trial

  1. Retrospective pre-implant symptom assessment and recall bias

Comment: “Recall bias can inflate post-implant increases; clarify direction/magnitude; STROBE form incorrectly marks bias as N/A.”

Response: Thank you for your suggested changes. The direction and potential impact of recall bias are explained in detail in the text. Where necessary, comparisons limited to clinical records were noted and highlighted in yellow.

Manuscript changes:

  • It is explained that recall bias may lead to under-recalling of pre-implant symptoms and thus over-recalling of post-implant increases.
  • The N/A error in the STROBE form was corrected; an explanation was added to the bias section.
  • Although access to records was limited, it was noted that clinical notes partially confirmed symptom categories in some patients.

  1. İstatistiksel raporlama – etki büyüklükleri, CI, çoklu karşılaştırmalar

Comment: “Add 95% CIs, describe familywise error/multiple comparisons control, clarify covariates.”

Response: Thank you for your suggested changes. All tables have been reorganized according to your suggestions.

Manuscript changes:

  • The 95% CI values for symptom changes were added to Table 3, and 4.
  • A subgroup analysis of symptom changes according to shock status was performed; however, since the analysis was exploratory and descriptive in nature, it was noted that no interaction test or formal hypothesis test was performed between subgroups, and this was highlighted in yellow.

  1. Repeated statements, restructuring of the discussion

Comment: “Fatigue statements repeated; consolidate paragraphs. Aromatherapy RCT paragraph should be moved into a dedicated subsection.”

Response: Thank you for your suggested changes. Paragraphs were merged and a separate “Future Directions” subsection was created for the aromatherapy RCT and highlighted in yellow.

Manuscript changes:

  • Physiological and psychological explanations related to fatigue were consolidated into a single summary paragraph.
  • The aromatherapy RCT (NCT07121855) was explained under the heading “Future Research” to ensure a more consistent flow of the text.

  1. Cultural coping methods – measurement validity

Comment: “How was ‘effective’ defined? Was the scale validated? Provide item wording.”

Response: Thank you for your suggested changes. We have added detailed explanations on this subject and highlighted them in yellow.

Manuscript changes:

  • The responses “effective/partially effective/ineffective” were measured using a single-item subjective rating scale; the original wording of the item was added to the Methods section.
  • Since this measurement was not validated, it was discussed that it was accepted at the level of face validity and that this created limitations.
  • It was suggested that a validated scale for cultural coping be added in future studies.

  1. Biophysical data – ICD shocks and device logs

Comment: “Missing objective device data (shocks/ATP). Provide or discuss implications.”

Response: Thank you for your suggested changes. We assessed the effect on symptom changes reported by patients by reaching the shock status from device records and performing subgroup analysis, highlighting the relevant sections in yellow.

Manuscript changes:

  • Subgroup analysis for patients who received and did not receive shock was added to the Results section because device records were accessible.
  • The Discussion section now includes a discussion of the possible reasons why no difference was observed in this study, despite widespread evidence that the shock experience increases symptom severity.

  1. Clarity of tables – scale explanations and inclusion of CI

Comment: “Add scale range explanations; clearer labeling; include CI.”

Response: All tables have been revised.

Manuscript changes:

  • Symptom score range (0–3) added to the methods section
  • 95% CI added to Table 3.
  • Headings clarified again.

  1. STROBE / CONSORT reporting

Comment: “Some items marked N/A incorrectly. Fix and justify.”

Response: The STROBE form was reviewed again and sections such as bias and subgroup analysis were corrected.

  1. Responses to the referee's questions

The following explanations were added to the article in Methods or Limitations.

  1. Recall bias verification:

  • Due to limited access to patients' clinical records, extensive validation could not be performed; however, symptom themes were reported to be consistent in a small subgroup.
  • The direction of recall bias (lower recall of pre-implant symptoms) was explained.

  1. “Effective/partially effective/ineffective” measurement:

  • The full text of the scale item was added to the Methods section.
  • It was stated that there was no validation.

  1. ICD therapy data (shock/ATP):

  • Device therapy data (shock status) was clearly specified;
  • Subgroup data was reported.

Reviewer 2 Report

Comments and Suggestions for Authors

The authors present a cross-sectional observational study of 142 ICD patients followed at a university arrhythmia clinic in eastern Türkiye (May–August 2022).

They retrospectively assessed pre-implant symptoms (by recall after implantation) and simultaneously assessed post-implant symptoms, focusing on cardiac (chest pain, palpitations, dizziness, syncope, dyspnea) and non-cardiac symptoms (fatigue, insomnia, fear, anxiety). They also examined non-pharmacological coping strategies (prayer, herbal tea, walking, etc.) and their perceived effectiveness, and explored the association between coping methods and socio-demographic/clinical factors.

Comments:

  1. Pre-implant symptoms were retrospectively assessed after implantation, based on patient recall. In reality, the “before vs after” comparisons cannot be interpreted as true longitudinal changes — they mix actual change with memory distortion, adaptation, and current emotional state.
    Suggestion: Emphasize this limitation more clearly both in the Methods (when describing data collection) and in the Discussion/Conclusions.
    Results and conclusions should be softened: e.g. “patients reported fewer cardiac symptoms after ICD implantation” rather than “ICD implantation reduced symptoms”, and explicitly state that pre-implant data are retrospective and self-reported.
  1. In the Data Collection / Symptom Experiences and Coping Methods section, explicitly state the coding (e.g. none = 0, sometimes = 1, frequently = 2, always = 3). Clarify whether the scale has been validated in ICD patients or adapted from an existing instrument, and if so, provide the reference.
  2. The use of the Wilcoxon test for pre–post comparisons of ordinal symptom scores is appropriate, but the text mixes means/SD (which are for continuous data) with a non-parametric test. Median (IQR) might be more appropriate for ordinal outcomes, or you can keep means but clearly justify.
  3. Correct the p-value for fear, and ensure consistency between text, tables, and conclusions.
  4. The increase in fatigue and anxiety after ICD implantation is one of the most interesting and clinically important findings. However, the discussion remains relatively general and sometimes repetitive. More explicitly discuss why fatigue and anxiety might increase even as cardiac symptoms improve (e.g. chronic HF, deconditioning, fear of shocks, role changes, overprotective family attitudes).
  5. Consider grouping coping methods into broader categories (e.g. religious/spiritual, physical activity, complementary/traditional) and briefly interpret them in that framework.
  6. Add 2–3 sentences in the Discussion explicitly defining the added value: e.g. “to our knowledge, this is one of the first quantitative studies describing symptom trajectory and culturally rooted coping behaviors (e.g. prayer) in ICD patients in Türkiye.
  7. In the limitations, you should explicitly state that these findings may not be generalizable to countries or cultures with different religious/spiritual profiles or healthcare structures.
Comments on the Quality of English Language

Overall, the English is understandable and generally good. A few minor issues of repetition and phrasing could be streamlined, especially in the Discussion, where some sentences and ideas are repeated (e.g. fatigue complicating daily life). A light language/editing pass would improve concision and flow.

Author Response

  1. Retrospective pre-implant symptom assessment & interpretation of before/after comparisons

Comment: Pre-implant symptoms were recalled retrospectively; before–after comparisons do not represent true longitudinal change. Soften conclusions and emphasize this limitation.

Response: Thank you for your suggested changes. We fully agree that retrospective recall limits the causal interpretation of pre-post differences. We revised and highlighted in yellow the Methods, Discussion, and Results sections to emphasize that pre-implant symptoms were self-reported retrospectively and therefore reflect both actual change and potential memory bias or emotional reappraisal.

Manuscript changes:

  • The Methods section clearly states that pre-implant symptoms were collected through retrospective self-reporting and that this is not a true longitudinal measurement.
  • The Discussion and Conclusion sections emphasize that pre-post differences cannot be interpreted as true changes, but only as “changes reported by patients.”
  • The concluding statements were softened from “ICD implantation reduced symptoms” to “patients reported fewer cardiac symptoms after ICD implantation.”

  1. Symptom scale coding & validation

Comment: Explicitly describe coding (0–3); state whether scale is validated or adapted.

Response: Thank you for your suggested changes. We clarified the coding structure (0 = none, 1 = sometimes, 2 = frequently, 3 = always) and added a sentence describing that the scale is not a validated ICD-specific instrument but an adapted symptom frequency measure with face-validity. We also highlighted the changes in yellow.

Manuscript changes:

  • The 0–3 coding was explicitly added to the “Data Collection / Symptom Experiences and Coping Methods” section.
  • It was noted that the scale has not been validated in ICD patients, that it was adapted from symptom frequency items in the existing literature, and that this is a limitation.

  1. Wilcoxon use vs. mean/SD reporting

Comment: Wilcoxon is appropriate for ordinal data, but means/SD are mixed with a non-parametric test; consider medians (IQR) or justify.

Response: Thank you for your suggested changes. To ensure comparability with previous ICD symptom studies, we retained the means/SD, but since we added the 95% confidence intervals for symptom change differences to the tables, we continued with the means/SD and also explained in the Methods section that the Wilcoxon test was used. We also highlighted the changes in yellow.

Manuscript changes:

  • In the Methods section, the rationale for choosing a non-parametric test (ordinal data) was explained, and it was noted that the use of mean/SD was retained for consistency with the literature.

  1. p-value correction

Comment: Correct the p-value for fear; ensure consistency across text and tables.

Response: Thank you for your suggested changes. The error has been corrected. All p-values have been cross-checked for consistency and highlighted in yellow.

Manuscript changes:

  • The p-value table and text related to the Fear variable have been corrected to ensure consistency.

  1. Fatigue and anxiety increase — deepen discussion

Comment: Discussion should more explicitly address why fatigue/anxiety might increase even when cardiac symptoms improve.

Response: Thank you for your suggested changes. We expanded the topic by including possible mechanisms such as progression of heart failure, loss of fitness, fear of shock, role changes, and overly protective family dynamics, and highlighted them in yellow.

Manuscript changes:

  • A more detailed paragraph explaining the increase in fatigue and anxiety was added to the discussion section.
  • Repetitive phrases were removed to make the text more fluent.

  1. Group coping methods into broader categories

Comment: Group coping methods (religious/spiritual, physical activity, complementary).

Response: Thank you for your suggested changes. In the Results section, we categorized coping behaviors into conceptual categories and discussed their cultural significance in Turkey in detail, highlighting them in yellow.

Manuscript changes:

  • Coping methods have been reorganized into three categories: religious/spiritual, physical activity, complementary/traditional practices.
  • Brief explanations regarding the cultural interpretation of these categories have been added to the discussion section.

  1. Clarify added value / novelty

Comment: Add 2–3 sentences explicitly defining added value (culturally rooted coping, Turkish ICD population).

Response: Thank you for your suggested changes. We expanded and highlighted the Discussion section in yellow to emphasize that this study is one of the first quantitative analyses of symptom progression and culturally embedded coping behaviors (e.g., prayer) among ICD recipients in Türkiye.

Manuscript changes:

  • New sentences summarizing the contribution of the study to the debate have been added: cultural context, Turkish sample, quantitative reporting of coping behaviors.

  1. Generalizability limitations

Comment: State that findings may not generalize to cultures with different religious/spiritual structures.

Response: Thank you for your suggested changes. We agree with this point and have clearly added it to the Limitations section and highlighted it in yellow.

Manuscript changes:

  • Added to the Limitations section that the study may be specific to societies that use religious/spiritual coping methods to a high degree and that its generalizability to other cultures is limited.

  1. English language quality

Comment: English is good but repetitive; streamline Discussion.

Response: Thank you for your suggested changes. We have edited to shorten the discussion, removed repetitions, revised our wording for clarity, and highlighted it in yellow.

Manuscript changes:

  • Repetitive sentences were removed from the discussion section and minor linguistic adjustments were made to the text.
  • Fluency was improved and unnecessary repetitions were eliminated.

Round 2

Reviewer 1 Report

Comments and Suggestions for Authors

Thank you for the revision, i have no further suggestion.

Reviewer 2 Report

Comments and Suggestions for Authors

The authors have addressed my concerns.